# Rapid Detection of Antimicrobial Resistance Genes in Critically Ill Children Using a Custom TaqMan Array Card

**DOI:** 10.3390/antibiotics12121701

**Published:** 2023-12-05

**Authors:** John A. Clark, Martin D. Curran, Theodore Gouliouris, Andrew Conway Morris, Rachel Bousfield, Vilas Navapurkar, Iain R. L. Kean, Esther Daubney, Deborah White, Stephen Baker, Nazima Pathan

**Affiliations:** 1Department of Paediatrics, University of Cambridge, Cambridge CB2 0QQ, UK; irlk2@cam.ac.uk (I.R.L.K.); esther.daubney@nhs.net (E.D.); deborah.white31@nhs.net (D.W.); np409@cam.ac.uk (N.P.); 2Cambridge University Hospitals NHS Foundation Trust, Cambridge CB2 0QQ, UK; theodore.gouliouris@nhs.net (T.G.); ac926@cam.ac.uk (A.C.M.); rachel.bousfield@nhs.net (R.B.); vilas.navapurkar@nhs.net (V.N.); 3Clinical Microbiology and Public Health Laboratory, United Kingdom Health Security Agency, Cambridge CB2 0QQ, UK; martin.curran@nhs.net; 4Division of Anaesthesia, Department of Medicine, University of Cambridge, Cambridge CB2 2QQ, UK; 5Division of Immunology, Department of Pathology, University of Cambridge, Cambridge CB2 1QP, UK; 6Cambridge Institute of Therapeutic Immunology and Infectious Disease, University of Cambridge, Cambridge CB2 0AW, UK; sgb47@cam.ac.uk

**Keywords:** rapid diagnostic tests, antibacterial agents, critical illness, child, respiratory system, antimicrobial resistance

## Abstract

Bacteria are identified in only 22% of critically ill children with respiratory infections treated with antimicrobial therapy. Once an organism is isolated, antimicrobial susceptibility results (phenotypic testing) can take another day. A rapid diagnostic test identifying antimicrobial resistance (AMR) genes could help clinicians make earlier, informed antimicrobial decisions. Here we aimed to validate a custom AMR gene TaqMan Array Card (AMR-TAC) for the first time and assess its feasibility as a screening tool in critically ill children. An AMR-TAC was developed using a combination of commercial and bespoke targets capable of detecting 23 AMR genes. This was validated using isolates with known phenotypic resistance. The card was then tested on lower respiratory tract and faecal samples obtained from mechanically ventilated children in a single-centre observational study of respiratory infection. There were 82 children with samples available, with a median age of 1.2 years. Major comorbidity was present in 29 (35%) children. A bacterial respiratory pathogen was identified in 13/82 (16%) of children, of which 4/13 (31%) had phenotypic AMR. One AMR gene was detected in 49/82 (60%), and multiple AMR genes were detected in 14/82 (17%) children. Most AMR gene detections were not associated with the identification of phenotypic AMR. AMR genes are commonly detected in samples collected from mechanically ventilated children with suspected respiratory infections. AMR-TAC may have a role as an adjunct test in selected children in whom there is a high suspicion of antimicrobial treatment failure.

## 1. Introduction

Antimicrobial-resistant lower respiratory tract infection (AMR-LRTI) was the leading cause of AMR-related deaths in 2019, responsible for >1.5 million deaths worldwide [1]. AMR-LRTI appears to be uncommon in critically ill children in high-income countries [2,3]; however, this may be due to under-detection. Microbiological cultures are low-yield in critically ill children, likely due to the early administration of broad-spectrum antimicrobial therapy [4]. Traditionally, antimicrobial susceptibility testing is performed by isolating an organism followed by phenotypic tests such as determining the minimum inhibitory concentration (MIC) against antimicrobials. Molecular methods are an alternative that can provide diagnostic information in the absence of a cultured organism. Rapid detection of AMR genes has the potential to provide earlier data to support the rationalisation of antimicrobial therapy, which is vital in preventing AMR.

Here we describe the validation and potential utility of a custom AMR-TAC that can detect up to 25 AMR genes which can confer β-lactam, carbapenem, fluoroquinolone, glycopeptide, and macrolide resistance. This is broader than the range of AMR genes incorporated in commercially available syndromic diagnostic tests for respiratory infection (Table 1) [5,6]. In addition, we report AMR gene carriage in mechanically ventilated children with suspected respiratory infections using this method for the first time using samples from a previously published cohort study [7]. Two body compartments were utilised to survey AMR gene carriage—the gastrointestinal tract and the lung. The gut has the greatest diversity and population of bacteria in the body, followed by the oral cavity [8,9]. Therefore, this provides the potential to obtain biomass-enriched samples. The gut is also a known reservoir for Gram-negative bacteria, particularly those with multi-drug resistance genes [10]. For these reasons, faecal samples are frequently used for AMR surveillance studies [9]. In contrast, lung samples have a lower biomass [11], and it is unclear whether this compartment could have a role in AMR surveillance and identification in mechanically ventilated children. A two-compartment approach was taken given the potential of horizontal gene transfer, which is the ability of bacterial species to transfer or acquire DNA allowing them to survive in hostile environments [12]. If the resistome of the gut is reflective of that of the respiratory tract, stool samples could be a non-invasive test for screening children for AMR. This is plausible given the respiratory tract of ICU patients can be colonised with gut bacteria given the presence of endotracheal tubes, translocation of bacteria via the bloodstream, and environmental cross-contamination.

The primary objectives of this study were as follows:Validate a custom AMR-TAC using isolates with known phenotypic and genotypic AMR.Identify the prevalence of AMR genes found in critically ill children with suspected LRTIs.

The secondary objectives of this study were as follows:Identify the correlation between the gastrointestinal and respiratory resistomes in critically ill children with suspected LRTIs.Describe the AMR genes found in critically ill children with suspected LRTIs who had AMR identified using conventional antimicrobial susceptibility testing.

## 2. Results

### 2.1. Antimicrobial Resistance Gene TaqMan Array Card Validation

All AMR gene assays passed validation with plasmid controls except for *bla*_CTX-M_ #2 (assay 23), which was negative despite VAP plasmid 2 containing this gene (Appendix A). The median standard deviation of Ct values on assays obtained across multiple AMR-TACs was 1.2 (range 0.3–2.3) (Appendix A); hence, there was a high degree of inter-assay reproducibility. The *bla*_CTX-M_ #2 gene (assay 23) was again not detected in these samples, and there was no detection of *bla*_CTX-M-2_ and *bla*_CTX-M-8/25_; however, these genes were unlikely to have been present in the original sample.

AMR-TAC had a 58/59 (98%) positive percentage agreement in detecting AMR genes incorporated in the AMR-TAC (Appendix A). Of the clinical samples used for validation, the presence of AMR genes or the phenotypic AMR profile was known in 59/60 samples. TAC fully corresponded to the known resistance of 39/59 (66%) of samples and partially corresponded to 20/59 (34%). Partial correspondence was due to the detection of additional AMR genes of uncertain significance and one missed detection of *bla*_OXA-1_. Two samples (3%) were known to be phenotypically resistant (samples 39 and 40); however, no AMR genes were identified. This may have been due to the presence of AMR genes, which did not have corresponding assays on the AMR-TAC or other resistance mechanisms.

### 2.2. Demographics

There were 82/100 children in the cohort that had a mini-BAL and/or faecal sample available for analysis. These children had a median age of 1.2 years (IQR 0.3–4.9 years), with the majority admitted to PICU for a primary respiratory problem (Table 2). There was 79/82 (96%) PICU survival, with longer lengths of stay in children with suspected HAP/VAP (*p* = 0.020).

At least one AMR gene was detected in 49/82 (60%) of children, and multiple AMR genes were detected in 14/82 (17%) of children (Figure 1). The most detected AMR gene was the erythromycin resistance methylase B (*ermB*) gene (*n* = 45; 54% children), followed by *mecA* (*n* = 14; 17%) and *mecC* (*n* = 2; 2%). There were single detections of *bla*_CTX-M-1_, *bla*_CTX-M-9_, *vanA*, and *vanB*.

There was a significantly greater proportion of AMR genes detected in faecal samples than mini-BAL samples (37/68 (54%) faecal samples, 22/81 (27%) mini-BAL samples, *p* < 0.001). There was a slight agreement between respiratory and faecal specimens in the 67 patients in which both samples were available for testing on the AMR-TAC (14% positive concordance, 87% negative concordance, κ = 0.187, CI_95_ 0.051–0.323) (Appendix A).

### 2.3. Respiratory Microbiology Results Compared to an Antimicrobial Resistance Gene TaqMan Array Card

A bacterial isolate was identified in 13/82 (16%) of lower respiratory (ETA or mini-BAL) cultures. Of these, 4/13 (31%) were phenotypically resistant to antimicrobials (Table 3). Although the *ermB* gene was detected in two of these patients’ respiratory secretions, bacterial isolates grown from these samples had no phenotypic macrolide resistance.

### 2.4. Antimicrobial Resistance Gene TaqMan Array Card Detections Correlating to Clinical Cases

There were two instances in which there was no growth on the first respiratory culture obtained as per protocol, but phenotypic AMR was detected in subsequent microbiology investigations correlating with AMR-TAC (Table 4).

Patient C060 was an 11-year-old with suspected VAP on a background of peritonitis secondary to a ruptured appendix. He received seven days of treatment with fluconazole and meropenem and one day of tigecycline. In the PICU, a multi-drug-resistant *E. faecium* was identified in ETA. The initial treatment was tigecycline and fluconazole. The *vanA* gene was detected in the respiratory sample.

Patient C094 was an 11-year-old with suspected HAP following orthopaedic surgery. She was at high risk of AMR due to recent hospitalisation and antimicrobial therapy in addition to a background of global developmental delay, a seizure disorder, and gastrostomy feeding. She needed mechanical ventilation due to acute respiratory distress syndrome and was treated with meropenem. Whilst her respiratory cultures did not identify a pathogen, extended-spectrum β-lactamase (ESBL) screening swabs were positive. Consistent with this finding, *bla*_CTX-M-1_ and *bla*_CTX-M_ were detected in stool.

## 3. Materials and Methods

### 3.1. Study Design and Population

This observational cohort study was designed to evaluate the use of an AMR-TAC in children with suspected respiratory infections. Details of the ethical approval and consent process have been previously published [13]. Patients were enrolled in the study between April 2020 and January 2022 in a 13-bed general PICU at Addenbrooke’s Hospital, Cambridge, England.

### 3.2. Eligibility Criteria

The inclusion criteria were as follows:The child was aged <18 years old;The child was receiving mechanical ventilation at the time of enrolment;The child was commencing or already receiving antimicrobial therapy to treat a suspected or confirmed LRTI.

The exclusion criteria were as follows:The patient had a non-survivable illness and was no longer on an active treatment pathway;The child was aged <37 weeks corrected gestation.

### 3.3. Non-Bronchoscopic Bronchoalveolar Lavage Sampling

Deep respiratory samples were obtained at the time of enrolment via non-bronchoscopic bronchoalveolar lavage (Mini-BAL). This is a technique in which saline is instilled at the level of the carina via the endotracheal tube and then aspirated into a sterile sputum trap. The saline lavage volume instilled was 1 mL/kg of patient weight to a maximum of 10 mL. Once aliquots of the sample had been obtained for clinical investigations, the remaining sample was stored at −80 °C until batch processing.

### 3.4. Faecal Sampling

Faecal samples were obtained by swabbing stool in the child’s nappy or via rectal swab. The swab was inserted into the anus to the end of the cotton tip. Swabs were then placed in a sterile container containing 1 mL DNA/RNA shield (Zymo Research, Irvine, CA, USA) and stored in a −20 °C freezer.

### 3.5. Nucleic Acid Extraction from Non-Bronchoscopic Bronchoalveolar Lavage Samples

Up to 750 μL of sample was added to a 2 mL microtube containing a mixture of 1.4 mm ceramic beads (Cat No. 13113-325, Qiagen, Hilden, Germany) with 750 μL of L6 buffer (50% guanidine thiocyanate, 20 mM EDTA, 1.3% Triton-X-100, Tris pH 7.25). A minimum of 100 μL was required for extraction and brought up to 750 μL with nuclease-free water in the case of low volume. The sample was then vortexed at 7000 rpm for one minute using a MagNA lyser (Roche, Basel, Switzerland) and spun at 13,000 rpm for one minute. Then, 400 μL of the sample was taken for extraction.

Samples were processed using an EZ1 virus mini kit (v 2.0) using an EZ1 advanced XL (Qiagen, Hilden, Germany) following standard methods to produce a 150 μL elution [14]. This could be completed with up to 14 samples per run, including an L6 buffer and RNase-free water negative control [14].

### 3.6. Nucleic Acid Extraction from Faecal Samples

The stored faecal specimens were thawed in batches and processed using a DNeasy^®^ PowerSoil^®^ Kit (Qiagen, Hilden, Germany) using standard methods. Here this protocol is briefly summarised. All the described reagents were included in the commercial kit. The sample was added to the PowerBead tube and vortexed. Then, 60 μL of solution C1 was added, and the tube was vortexed for 10 min. Tubes were then vortexed at 10,000× *g* for 30 s. The supernatant was transferred to a clean 2 mL collection tube to which 250 μL of solution C2 was added, and the sample was vortexed for 5 s. It was then incubated for 5 min at 2–8 °C. The tubes were vortexed for 1 min at 10,000× *g*, and 600 μL of the supernatant was transferred to a clean 2 mL collection tube. Then, 200 μL of solution C3 was added, and the tube was vortexed and then incubated for 5 min at 2–8 °C. Tubes were then vortexed for 1 min at 10,000× *g.* Avoiding the pellet, 750 μL was transferred to a clean 2 mL collection tube. Solution C4 was shaken, and 1200 μL was added to the supernatant. The tube was vortexed for 5 s. Then, 675 μL was loaded onto an MB spin column and centrifuged at 10,000× *g* for one minute. The spin column was then placed in a clean 2 mL collection tube, and 100 μL of Solution C6 was added to the centre of the filter membrane. The tube was then centrifuged for 30 s at 10,000× *g*. Extracted samples were then stored at −80 °C.

### 3.7. Nucleic Acid Extraction from Raw Sewage Samples

Sewage samples were extracted according to a validated protocol [15], briefly summarised here. Sample bags were sprayed with 80% ethanol inside a class II microbiological safety cabinet. The sample bag was then opened, and the tube was sprayed with 80% ethanol. Then, 500 μL of 100% ethanol was added to the sample and incubated for 10 min. Then, 400 μL lysis buffer was added, and 600 μL of the sample was added to a silica spin column. This was centrifuged at 15,000 rpm for 30 s. The follow-through was discarded, the remaining 600 μL of the sample was added to the spin column, and the tube was centrifuged at 15,000 rpm for 30 s. The follow-through was discarded. After the addition of 500 μL of wash buffer one (1 M guanidine thiocyanate in 25 mM Tris-HCl, with 10% ethanol), the column was centrifuged for 30 s at 15,000 rpm. The follow-through was discarded. Then, 500 μL of wash buffer two (25 mM Tris-HCl buffer with 70% ethanol) was added to the spin column and centrifuged for 30 s at 15,000 rpm. The follow-through was discarded. Then, 500 μL of wash buffer two was added to the spin column and centrifuged for 2 min at 15,000 rpm. In a new collection tube, the sample was spun for 1 min at 15,000 rpm. Then, 100 μL of nuclease-free water was added to the spin column filter and left for 1 min. The sample was then spun for 15,000 rpm. The spin column was discarded, and the elution was stored at −80 °C.

### 3.8. Antimicrobial Resistance Gene TaqMan Array Card

A custom AMR-TAC was designed using a combination of commercial and in-house sequences to detect up to 25 AMR genes (Figure 2). The genes were selected to identify resistance to a wide range of antimicrobials. One lane of the array was required per sample. Most AMR gene assays were in duplicate to help identify potential false positives. An 18S rRNA gene target was included for quality control. There were ten custom assays on the card in addition to a custom MS2 bacteriophage control; names are provided for the commercial assays in which the primer sequences are not disclosed (Appendix A). The MecC assay detected both *mecA* and *mecC*; therefore, it was necessary to review it in conjunction with the MecA assay to determine which gene was present.

For each sample, 25 μL of total nucleic acid was added to 25 μL of TaqMan Fast Virus 1-step master mix (ThermoFisher, Foster City, CA, USA) and 50 μL of RNase-free water. A total of 50 μL was then added to the array. The RT-qPCR was completed on a QuantStudio 7 Flex (ThermoFisher) according to the following validated protocol: 50 °C for 5 min, 95 °C for 20 s, 45 cycles of 95 °C for 1 s, 60 °C for 20 s [16]. qPCR Ct values with clear amplification curves were reported.

A positive detection on the TAC was defined as follows:For AMR genes with one target on the TAC, Ct value ≤ 32;For AMR genes with ≥2 targets on the TAC, either of the following:
(a)At least one target, Ct ≤ 32;(b)At least two targets had Ct < 34.

The Ct cut-offs were selected based on a study of the use of AMR-TAC configured for *E. coli* isolates and stool specimens in which Ct ≤ 32 was optimal in positive/negative characterisation of samples compared to routine investigations through analysis with receiver operating curves (ROCs) [17]. A slightly higher Ct < 34 was permitted if present in at least two targets, given the low quantity of bacterial DNA expected to be present in mini-BAL samples.

The AMR-TAC used in this study was validated in three stages. Firstly, three synthetic positive control plasmids containing sequences for the in-house assays were tested in addition to a NATrol Pneumonia Verification Panel (NPVP) (NATPPQ-Bio and NATPPA-Bio, ZeptoMetrix, Buffalo, NY, USA). The plasmids were diluted tenfold to a range of 10^−6^. Testing was undertaken on raw sewage samples to determine the reproducibility of results between different AMR-TACs. Extracted samples were then tested on seven AMR-TACs undertaken at different times. Finally, extracted nucleic acid samples from 60 microbiology isolates with a range of AMR profiles were tested on the AMR-TAC. This included samples that had whole genome sequence data available, those that had undergone PCR for specific AMR genes, and those that had phenotypic resistance. FastQ files of sequenced National Collection of Type Cultures (NCTC) strains were downloaded from the European Nucleotide Archive (ENA). These files were then read in ResFinder 4.0 [18], with all identified AMR genes compared to AMR-TAC.

### 3.9. Conventional Respiratory Pathogen Testing

All routine microbiology testing was completed by qualified UK Health Security Agency (UKHSA) biomedical scientists according to laboratory standard operating procedures. For patients that were not immunocompromised, standard media were used. The sample was inoculated on chocolate agar and incubated at 35–37 °C supplemented with 5–10% CO_2_ [19]. If the patient was immunocompromised, MacConkey agar and Mannitol salt/chromogenic agar were used with the sample incubated in air [19]. Significant growth constituted >10^4^ cfu/mL on Mini-BAL samples or >10^5^ cfu/mL on ETA. Bacterial organisms were identified to the species or genus level using matrix-assisted laser desorption/ionisation time-of-flight (MALDI-TOF) mass spectrometry (Bruker Daltonics, Coventry, UK). Antimicrobial susceptibility testing was performed using disc diffusion following European Committee on Antimicrobial Susceptibility Testing (EUCAST) guidelines [20].

### 3.10. Data Collection

Data were obtained from the electronic medical record and recorded in an electronic research database, RedCAP, hosted by the University of Cambridge [21]. Demographic information (age at admission, admission source, diagnosis and comorbidities, AMR risk factors, days of therapy, and duration of admission), all routine microbiology investigations, antimicrobial prescription data, and Paediatric Index of Mortality 3 scores [22] were collected. Treatments and duration of stay are reported as days free of treatment 28 days following admission to PICU. This is a standard composite measure that captures the impact of mortality, with days following death considered unliberated from treatment for the purposes of the calculation [23].

### 3.11. Statistical Analysis

Statistical analysis was undertaken in R [24]. Demographic data were reported with mean and standard deviation, unless the distribution of data was skewed, in which case median and interquartile range were used. Demographic features were compared using the Mann–Whitney U test for non-parametric data and the Student’s *t*-test for normally distributed data. Proportions were compared using the chi-square test for independence. Concordance of AMR gene detections on faecal and respiratory specimens was reported both with concordance (%) and Cohen’s kappa. This was categorised using Altman’s description of the agreement [25]. Figures were created using BioRender.com, accessed 13 February 2023.

## 4. Discussion

Here we report the use of a custom AMR-TAC in critically ill children for the first time.

All assays, except for one of the two for *bla*_CTX-M_, were reliable. During validation, the AMR-TAC had 98% positive percentage agreement in identifying genes corresponding to the phenotypic resistance identified in microbiological culture susceptibility testing. We found that 60% of mechanically ventilated children with suspected LRTIs had at least one AMR gene present in mini-BAL and/or faecal swab samples.

The most prevalent gene identified in children with suspected LRTIs was *ermB*, which can confer macrolide–lincosamide–streptogramin B (MLS_B_) resistance, similar to a previous study that used the Unyvero P50 pneumonia cartridge (Curetis, Holzgerlingen, Germany) [5]. This may be explained by its high rates of carriage by common bacterial respiratory pathogens, including *S. pneumoniae* [26,27], *S. aureus* [28,29], and *H. influenzae* [30]. Children with erythromycin-resistant *S. pneumoniae* have significantly higher rates of carriage of *ermB* than those with erythromycin-sensitive *S. pneumoniae* (83.8% carriage of *ermB* in erythromycin-resistant isolates versus 17.8% carriage in sensitive isolates, *p* < 0.001) [26]. Macrolides are recommended in combination with β-lactams for the treatment of severe pneumonia in children or if there is suspicion of *Mycoplasma* or *Chlamydia* pneumonia [31]. The rates of macrolide-resistant *S. pneumoniae* are increasing, and macrolide-resistant *S. pneumoniae* is found in up to 40% of isolates obtained from adults in the United States [32]. It is, therefore, a potential concern that this treatment has the potential to be rendered ineffective. The *ermB* gene was recently ranked among the most concerning AMR genes, given it is present in a broad range of known pathogens, is found enriched in the human environment, and is contained in genetic material easily transferred between bacteria [33]. It is of note that we found this to be a highly prevalent gene in mechanically ventilated children.

ESBL genes were only found in 2/82 (2%) of our cohort. This is similar to the rate (2.8%) of ESBL found in faecal specimens of children admitted to the Johns Hopkins PICU, Baltimore, MD; of these ESBL cases, 23/24 (96%) had *bla*_CTX-M_ genes identified [34]. Genes linked with ESBLs and carbapenemases, including *bla*_CTX-M-15_, *bla*_NDM_, *bla*_KPC_, and *bla*_OXA-48_, have been found in up to 56.1% of neonates with suspected sepsis in low- and middle-income countries [35]. This is likely due to broad-spectrum antimicrobials being frequently prescribed in these infants in the setting of local infection control practices and a high rate of AMR in the local population. The AMR-TAC may have the greatest utility in populations with a high prevalence of AMR colonisation or infection but limited resources to screen for a broad range of causes of AMR. There is a greater abundance of AMR genes at six weeks of age than in young children and adults, which may be related to the shift in the microbiome composition [36]. This highlights that the local epidemiology, antimicrobial use, and the age of patients must be considered in AMR gene detection studies.

During AMR-TAC validation, a broad range of AMR genes were detected in hospital sewage. The genes present could confer resistance to all five classes of antimicrobials to which resistance genes are sought by the AMR-TAC. Previous studies have shown that AMR in wastewater has a high level of agreement with AMR in human populations, confirmed with genotypic and phenotypic testing (0.85 concordance, CI_95_ 0.80–0.89) [37]. The AMR-TAC could be used as a population-level AMR surveillance tool, with the advantage that the method would not depend on the ability to culture all the isolates that may grow from the sample [38].

The AMR-TAC had a more comprehensive range of AMR genes than those incorporated in commercial LRTI diagnostic assays. In critically ill adults, the Unyvero HPN/P55 pneumonia cartridge has a sensitivity of 18.8% and a specificity of 94.4% for the detection of AMR [39]. The former P50 cartridge, which incorporated 18 resistance markers, identified at least one AMR gene in 64/90 (71%) respiratory specimens obtained from adults hospitalised with pneumonia [40]. Concordance between phenotypic resistance and AMR genes found on BioFire FilmArray Pneumonia Panel (FPP) has been highly variable in adult pneumonia studies [41,42,43,44,45]. Given difficulties interpreting the significance of these genes and discrepancies with microbiological culture susceptibility data, there was no guidance provided to clinicians in the prescribing protocol of the recent INHALE trial [46]. This is despite the FPP only reporting AMR genes known to be present in bacteria it co-detects [47]. We identified a higher prevalence of AMR gene detection compared to a study of the FPP undertaken on BAL samples of hospitalised patients. In that investigation, *mecA/mecC* and MREJ were found in 43/666 (6.5%) of samples, *bla*_CTX-M_ in 7/666 (1.1%), *bla*_KPC_ in 2 (0.3%), and *bla*_NDM_ in 1 (0.2%) [47].

In adults with LRTIs, metagenomic next-generation sequencing (mNGS) is associated with 70% sensitivity (CI_95_ 47–87%) and 95% specificity (CI_95_ 85–99%) for the detection of AMR in Gram-positive organisms and 100% sensitivity (CI_95_ 87–100%) and 64% specificity (CI_95_ 48–78%) in Gram-negative organisms [48]. mNGS is less sensitive than qPCR in measuring AMR gene abundances and requires a human DNA depletion step for most sample types aside from faeces [49]. Whilst mNGS can identify an unlimited number of AMR genes, so long as they are present in a reference database, it is currently not a practical approach due to the processing involved and cost [49].

The strengths of this study were the integration of clinical and microbiological data and the inclusion of a broad range of AMR genes on the AMR-TAC. The validation of the array before testing the cohort of PICU patients demonstrated that, except for one of the *bla*_CTX-M_ assays, the array was highly sensitive for the detection of AMR genes. An alternative probe could be selected for the *bla*_CTX-M_ gene in future studies. The study design was unique in that there have been no similar screening studies of a wide range of AMR genes in PICUs, to our knowledge.

There were several limitations. Firstly, not all children enrolled in the project had a mini-BAL and faecal sample obtained or a sufficient sample for undergoing AMR-TAC testing after routine investigations. Secondly, it is not possible to be certain as to the significance of detections in the validation of the AMR-TAC, where this was not associated with phenotypic resistance. These genes may have been present but not expressed, or they may have been present in other colonising bacteria not isolated on culture. Future studies may therefore consider whole genome sequencing as an additional validation tool to determine gene presence. Such a study should consider concurrently evaluating the presence of regulatory genes, which can impact the MIC of bacteria [50]. Finally, the optimal Ct value cut-offs for this array require further research. The Ct value obtained does not necessarily correlate with the number of genes present in the original bacteria in which they were found. Therefore, the degree of the potential resistance conferred by a low or high Ct value is difficult to determine.

This institution’s PICU has a low incidence of AMR [3]. In the PICU, children at the highest risk of multi-drug-resistant organisms (MDROs) are those who have coma, are receiving parenteral nutrition, have two or more antimicrobials prescribed, and/or are receiving five or more days of mechanical ventilation [51]. These children may be the best target for an AMR screening study using the AMR-TAC. Importantly, the prevalence of AMR may be much higher than previously identified in PICUs due to the limitations of microbiological cultures. Although only thirteen children had a positive respiratory microbiological culture in this cohort, 4/13 (31%) were phenotypically resistant to at least one class of antimicrobials. The AMR-TAC may help reveal underlying AMR, given the limitations of routine investigations. Only two children were identified who had AMR gene detections, confirmed on routine investigations, that may have impacted clinical management. Given the cost–benefit of the investigation, we suggest future studies focus on a subset of children with antimicrobial treatment failure. Testing across several geographical locations could also help better understand the utility of the array, particularly in human populations and environments with high AMR prevalence. These regions may need to select alternative gene targets based on their localised patterns of resistance. Additional healthy control patients would also serve as a helpful indicator of the baseline presence of AMR genes, given the cohort had a high level of recent hospitalisation and major comorbidity. An environmental surveillance study could also identify vectors for AMR in the PICU environment.

## 5. Conclusions

AMR genes are commonly detected in samples collected from ventilated children with suspected respiratory infections. The AMR-TAC does not have a role in replacing antimicrobial susceptibility testing of microbiology isolates, given this method does not identify the microorganism that contains the genes, determine if the genes are expressed, or provide antimicrobial breakpoints. The AMR-TAC has a potential role as an adjunct test in select children for whom there is a high level of suspicion of antimicrobial treatment failure combined with an absence of antimicrobial culture and susceptibility data to direct prescribing decisions.

## Figures and Tables

**Figure 1 antibiotics-12-01701-f001:**
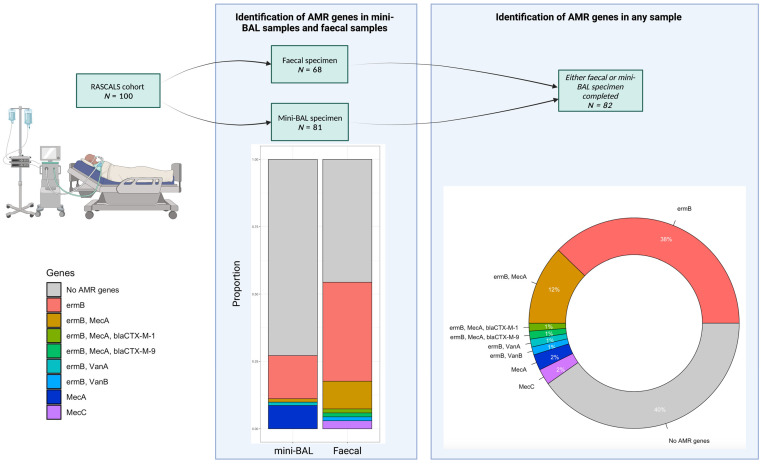
Antimicrobial resistance genes detected in critically ill children with suspected lower respiratory tract infection. This figure represents antimicrobial resistance (AMR) genes identified using a custom antimicrobial resistance gene TaqMan array card. AMR genes were detected in a greater proportion of faecal samples than non-bronchoscopic bronchoalveolar lavage (mini-BAL) samples. The most commonly identified AMR gene in faecal and mini-BAL samples was *ermB*. If expressed, this gene can confer resistance to macrolides. Figure created at BioRender.com (accessed on 13 February 2023).

**Figure 2 antibiotics-12-01701-f002:**
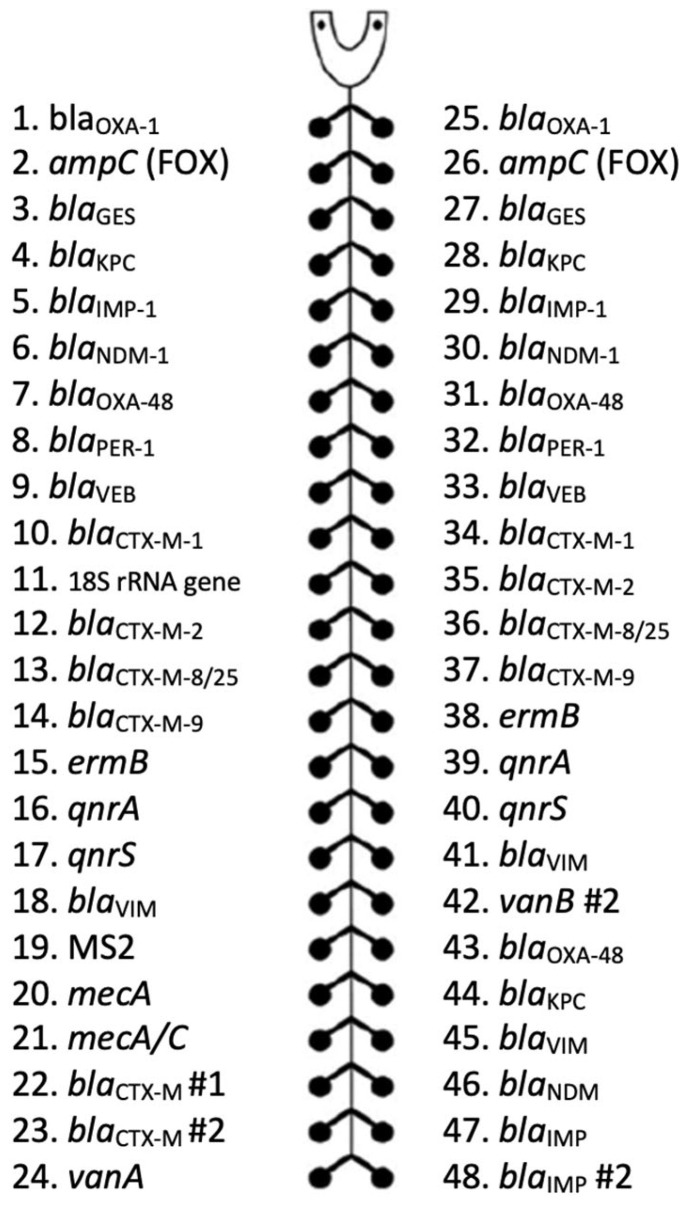
Configuration of a custom antimicrobial resistance gene TaqMan array card. This figure represents one lane of the antimicrobial resistance gene TaqMan array card (AMR-TAC). The card contains eight lanes with each configuration. The assays can detect genes conferring resistance to β-lactams, carbapenems, fluoroquinolones, glycopeptides, and macrolides.

**Table 1 antibiotics-12-01701-t001:** Antimicrobial resistance genes incorporated in a custom TaqMan array card compared to commercially available tests.

Antimicrobial Class	Resistance Target	BioFire Pneumonia Panel *N* = 7	Unyvero Pneumonia Cartridge *N* = 17	Custom AMR-TAC *N* = 25
β-lactamase	*AmpC* (FOX)			√
	*bla* _CTX-M_	√	√	√
	*bla* _CTX-M-1_			√
	*bla* _CTX-M-2_			√
	*bla* _CTX-M-8/25_			√
	*bla* _CTX-M-9_			√
	*bla* _OXA-1_			√
	*mecA*		√	√
	*mecC*		√	√
	*mecA*/*mecC*/MREJ	√		
	*bla* _PER-1_			√
	*bla* _SHV_		√	
	*bla* _TEM_		√	
	*bla* _VEB_			√
	*bla* _VIM_	√	√	√
Carbapenemase	*bla* _NDM_	√	√	√
	*bla* _NDM-1_			√
	*bla* _GES_			√
	*bla* _IMP_	√	√	√
	*bla* _IMP-1_			√
	*bla* _IMP-2_			√
	*bla* _KPC_	√	√	√
	*bla* _OXA-23_		√	
	*bla* _OXA-24/40_		√	
	*bla* _OXA-48- like_	√		
	*bla* _OXA-48_		√	√
	*bla* _OXA-58_		√	
Fluoroquinolone	*gyrA83*		√	
	*gyrA87*		√	
	*qnrA*			√
	*qnrS*			√
Glycopeptide	*vanA*			√
	*vanB*			√
Macrolide	*ermB*		√	√
Sulphonamide	*sul1*		√	

AMR-TAC: antimicrobial resistance gene TaqMan array card.

**Table 2 antibiotics-12-01701-t002:** Demographics of mechanically ventilated children screened for antimicrobial resistance.

Variable	Total	Suspected CAP	Suspected HAP/VAP	*p*-Value
	*N* = 82	*N* = 66	*N* = 16	
Demographics				
Age (years)—median (IQR)	1.2 (0.3–4.9)	1.0 (0.2–4.2)	2.8 (0.6–9.8)	0.124 ^a^
Sex (male)—*n* (%)	52 (63)	39 (59)	13 (82)	0.148 ^b^
Weight (kilograms)—median (IQR)	10.2 (5.3–18.0)	9.5 (5.0–17.3)	12.8 (6.6–26.7)	0.139 ^a^
Significant comorbidity—*n* (%)	29 (35)	21 (32)	8 (50)	0.172 ^b^
PIM3 score—median (IQR)	2.2 (0.5–4.6)	0.8 (0.5–4.6)	3.6 (2.1–5.0)	0.041 ^a^
Primary diagnostic category—*n* (%)				
Respiratory	52 (63)	47 (71)	5 (31)	0.003 ^b^
Neurological	9 (11)	8 (12)	1 (6)	0.094 ^b^
Cardiovascular	4 (5)	2 (3)	2 (13)	0.115 ^b^
Trauma	4 (5)	3 (5)	1 (6)	0.776 ^b^
Post-operative care	6 (7)	1 (2)	5 (31)	<0.001 ^b^
Other	7 (9)	5 (8)	2 (13)	0.527 ^b^
Risk factors for AMR—*n* (%)				
Home respiratory support (any)	6 (7)	6 (9)	0	0.210 ^b^
Tracheostomy	1 (1)	0	1 (6)	0.041 ^b^
NG feeding/gastrostomy	10 (12)	8 (12)	2 (13)	0.967 ^b^
Hospital admission within the last three months	40 (49)	35 (53)	5 (31)	0.118 ^b^
PICU admission within the last three months	15 (18)	11 (17)	4 (25)	0.036 ^b^
Previous mechanical ventilation	30 (37)	23 (35)	7 (44)	0.507 ^b^
Regular steroids	4 (5)	4 (6)	0	0.313 ^b^
Neutropaenia	1 (1)	1 (2)	0	0.620 ^b^
Malignancy	2 (2)	2 (3)	0	0.481 ^b^
Asplenia	1 (1)	1 (2)	0	0.620 ^b^
Known AMR—*n* (%)				
CPE	1 (1)	1 (2)	0	0.620 ^b^
ESBL	2 (2)	0	2 (13)	0.004 ^b^
None	79 (96)	65 (98)	14 (88)	0.036 ^b^
Days free of treatment at 28 days—mean (SD)				
Antimicrobial therapy	20.0 (7.4)	20.1 (7.3)	16.1 (8.2)	0.056 ^c^
Mechanical ventilation	19.3 (6.6)	19.8 (6.6)	15.8 (7.6)	0.064 ^c^
Inotropes	26.5 (4.3)	26.4 (4.7)	26.9 (1.5)	0.442 ^c^
PICU admission	17.4 (7.5)	18.1 (7.7)	13.3 (6.8)	0.020 ^c^
Survival to hospital discharge—*n* (%)	78 (95)	64 (97)	14 (88)	0.115 ^b^

^a^ = Mann–Whitney U test; ^b^ = chi-square test for independence; ^c^ = Student’s *t*-test for equality of means; AMR: antimicrobial resistance; CAP: community-acquired pneumonia; CPE: Carbapenemase-Producing *Enterobacteriaceae;* ESBL: extended-spectrum β-lactamase; HAP: hospital-acquired pneumonia; IQR: interquartile range; LRTI: lower respiratory tract infection; NG: nasogastric; PICU: paediatric intensive care unit; SD: standard deviation; VAP: ventilator-associated pneumonia.

**Table 3 antibiotics-12-01701-t003:** Antimicrobial resistance identified on culture compared to antimicrobial resistance gene TaqMan array card.

Study ID	Lower Respiratory Culture Result	Phenotypic Resistance	AMR-TAC Result, Ct Value(s)	
			Respiratory	Faecal
C008	*Pseudomonas aeruginosa*	Ciprofloxacin	Nil	N/A
C046	*Staphylococcus aureus*	Fusidic acid	Nil	N/A
C048	*Morganella morganii*	Amoxicillin/clavulanate Ampicillin/amoxicillin	*ermB* 28/28	Nil
C067	*Stenotrophomonas maltophilia*	Amoxicillin/clavulanate Ampicillin/amoxicillin	Nil	*ermB* 33/33

AMR-TAC: antimicrobial resistance gene TaqMan array card; Ct: cycle threshold; N/A: no sample available.

**Table 4 antibiotics-12-01701-t004:** Antimicrobial resistance gene detections on TaqMan array card correlating to phenotypic resistance.

Study ID	Organisms Identified on TAC (Ct Value(s))	Microbiological Culture Results	AMR-TAC Results (Ct)	
			Respiratory	Faecal
C060	Nil *	ETA: *E. faecium* Ampicillin/amoxicillin (R) Teicoplanin (R) Vancomycin (R) Peritoneal fluid: *E. faecium* Ampicillin/amoxicillin (R) Daptomycin (R) 2 mg/L Teicoplanin (R) Vancomycin (R)	*ermB* 25/25 *vanA* 27	N/A
C094	*Streptococcus* spp.: 29/27	ESBL screening: positive (multiple time points)	Nil	*ermB* 22/22 *bla*_CTX-M-1_ 26/26 *bla*_CTX-M_ 30 *mecA* 27/29

AMR-TAC: antimicrobial resistance gene TaqMan array card; Ct: cycle threshold ESBL: Extended-spectrum β-lactamase; ETA: endotracheal tube aspirate; R: full resistance; * there was no target for *E. faecium* on diagnostic TAC at the time this sample was tested.

## Data Availability

With the exception of potentially identifying information, study data are available at the Open Science Framework: Antimicrobial resistance gene TaqMan array card evaluation, DOI 10.17605/OSF.IO/89YMS.

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
