# Peer review of "Rapid Detection of Antimicrobial Resistance Genes in Critically Ill Children Using a Custom TaqMan Array Card"

_antibiotics, 2023, doi:10.3390/antibiotics12121701_

Round 1

Reviewer 1 Report

Comments and Suggestions for Authors

The purpose of this paper is to “describe the validation and potential utility of a custom AMR-TAC”.  The usual methodology in this cases is by comparing with a “gold standard” method or by using known isolates. In this paper neither is done. The authors used this under validation method to detect resistance genes in clinical samples, having an equation with two unknowns: the method to be validated and unknown samples. In that respect I  believe that the paper has serious methodological issues and that their results are not valid.  

minor comment: "The child was aged < 18 years old".  A child over 18 years is not a child

Author Response

  1. The purpose of this paper is to “describe the validation and potential utility of a custom AMR-TAC”. The usual methodology in this cases is by comparing with a “gold standard” method or by using known isolates. In this paper neither is done. The authors used this under validation method to detect resistance genes in clinical samples, having an equation with two unknowns: the method to be validated and unknown samples. In that respect I believe that the paper has serious methodological issues and that their results are not valid. This paper undertook validation of the AMR gene array using isolates with known phenotypic and genotypic resistance. This is included in the supplemental materials (table S4) and has been performed. After validation, the array was utilised on clinical samples and compared to the gold standard of antimicrobial susceptibility testing on microbiological culture.
  2. 2. minor comment: "The child was aged < 18 years old". A child over 18 years is not a child. The inclusion criteria specify that children are considered those less than 18 years old for clarity. In some countries such as the United States, paediatricians treat patients up to the age of 21 years.

Reviewer 2 Report

Comments and Suggestions for Authors

The article title” Rapid detection of antimicrobial resistance genes in critically ill children using a custom TaqMan array card” is very interesting. It’s about the method to detected wide range of AMR gene in critically ill children in order to help the efficient prescription treatment for the patient.

Anyway, there are some unclarity points and need some editing, which are

1.      The reason of choosing these panel of the AMR genes. If that involved the frequency detection of these AMR genes?

2.      Why you select to detect the AMR genes in feces. Although it non-invasive but is feces will represent as a good sample of the respiratory tract infection?

3.      In Method, could it be indicated the method of Antimicrobial resistance gene TaqMan array card in article not in the supplement data

4.      In Method, in 2.5 at the end of paragraph “with the MecA assay rto determine” should be “with the MecA assay to determine”

5.      If the pattern AMR genes in antibiotic class different according to the geographic area? If yes, please add this information in the discussion part so the application of this AMR-TAC would be considered in this point.

6.      About the validation step, you found out that blaCTX-M #2 (assay 23) was negative even the VAP plasmid 2 containing this gene. Could you please explain or suggest the resolving method?

7.      In discussion, page 10 “…prescribing protocol of the recent INHALE trial.” What is INHALE representing for?

8.      In case of in concordance of the genotypic of AMR genes and phenotypic culture-based assay, what do you suggest on this? Should that gene is the good represent of that antibiotic class.

Comments on the Quality of English Language

All content is well written, concise and easy to understand, anyway just one wrong typing should be edited.

Author Response

  1. The reason of choosing these panel of the AMR genes. If that involved the frequency detection of these AMR genes

The panel was selected based on several factors. Firstly, we wished to identify resistance across a wide range of commonly used antimicrobial agents in the PICU. The targets were selected based on association with high prevalence of clinical resistance, as per the Comprehensive Antibiotic Resistance Database (CARD).[1] A combination of gene targets was used, with some sequences from the manufacturer and others developed in-house based on sequencing performed on local pathogens. We integrated these targets to identify problematic resistance patterns identified in our region and country. For example, a high rate of VanA and VanB related vancomycin resistance has been identified in our institution,[2] hence these targets were incorporated. In England, the most common carbapenemase producing gram negative enzymes are blaOXA-48-like (34.9%) blaNDM (28.6%) and blaKPC (28.3%), hence inclusion of assays capable of these detections.[3]

Finally, targets were integrated that have already been included on commercially available respiratory diagnostic arrays. These include blaCTX-M, MecA, MecC, blaVIM, blaNDM, blaIMP, blaKPC, blaOXA-48 and ermB.[4,5] These arrays, unlike the TAC are not able to be customised as in the present study and incorporate a narrower range of gene targets.

  1. Why you select to detect the AMR genes in feces. Although it non-invasive but is feces will represent as a good sample of the respiratory tract infection?

Additional paragraph added to introduction – the gut is often used for microbiome studies as it is a high biomass compartment. If the gut resistome was reflective of the respiratory resistome it could potentially be used as a non-invasive collection method. The correspondence of results between lung and faecal samples was therefore evaluated. 

  1. In Method, could it be indicated the method ofAntimicrobial resistance gene TaqMan array card in article not in the supplement data

Additional details regarding the study methods have been moved from the supplemental materials to the manuscript as per reviewer suggestion.

  1. In Method, in 2.5 at the end of paragraph “with the MecA assayrto determine” should be “with the MecA assay to determine”

Correction made accordingly

  1. If the pattern AMR genes in antibiotic class different according to the geographic area? If yes, please add this information in the discussion part so the application of this AMR-TAC would be considered in this point.

Additional comment added to discussion – “Testing across several geographical locations, could also help better understand the utility of the array, particularly in human populations and environments with high AMR prevalence. These regions may need to select alternative gene targets based on their localised patterns of resistance”

  1. About the validation step, you found out that blaCTX-M #2 (assay 23) was negative even the VAP plasmid 2 containing this gene. Could you please explain or suggest the resolving method?

It is likely that this assay failed due to the MGB probe sequence (unable to anneal efficiently at 60 °C) used in this particular assay. An alternative probe sequence could be assessed first and then incorporated into assay 23 in future studies and this has been suggested in the discussion.

  1. In discussion, page 10 “…prescribing protocol of the recent INHALE trial.” What is INHALE representing for?

The INHALE study (although capitalised in the scientific literature), to our knowledge, does not appear to be an abbreviation. The study protocol paper, referenced in the present manuscript, described it as ‘the impact of using FilmArray Pneumonia Panel molecular diagnostics for hospital-acquired and ventilator-associated pneumonia on antimicrobial stewardship and patient outcomes in UK Critical Care

  1. In case of in concordance of the genotypic of AMR genes and phenotypic culture-based assay, what do you suggest on this? Should that gene is the good represent of that antibiotic class.

This is an interesting question but is difficult to answer with the present study. It is likely that the higher rate of AMR gene carriage, compared to phenotypic AMR in the study is due to these genes not being expressed. The present study was experimental and hypothesis generating, and future studies could undertake shotgun metagenomic sequencing to identify all AMR genes present in the cohort. This could allow the total AMR gene expression to be reported, but this would be best undertaken on a larger, multi-centre cohort.

Reviewer 3 Report

Comments and Suggestions for Authors

Comments for authors

1.       Abstract: remove the words: “Background:”, “Methods”, “Results:”, and “Conclusions:”

2.       Keywords: remove “-“ from antibacterial agents and add “antimicrobial resistance”

3.       The objectives of this study were not clear. Kindly more provide in the last paragraph of the Introduction section.

4.       What is the reason why the 18s rRNA gene of eukaryotic cells was used for quality control? Why not 16s rRNA from archaea and bacteria? Is this method directly targeting AMR genes in bacteria in the samples?

5.       The methods provided in the main text were inadequate. All must be moved from the supplementary file to the main text.

6.       If the microbiology-based technique was also processed. Kindly revise as above mentioned.

Author Response

  1. Abstract: remove the words: “Background:”, “Methods”, “Results:”,and “Conclusions:”

Change made accordingly

  1. Keywords: remove “-“ from antibacterial agents and add “antimicrobial resistance”

Change made accordingly

  1. The objectives of this study were not clear. Kindly more provide in the last paragraph of the Introduction section.

Objectives have been added as requested

Primary objectives

  1. Validate a custom AMR-TAC using isolates with known phenotypic and genotypic AMR.
  2. Identify the prevalence of AMR genes found in critically ill children with suspected LRTI.

Secondary objectives

  1. Identify the correlation between the gastrointestinal and respiratory resistomes in critically ill children with suspected LRTI.
  2. Describe the AMR genes found in critically ill children with suspected LRTI who had AMR identified using conventional antimicrobial susceptibility testing.

  1. What is the reason why the 18s rRNA gene of eukaryotic cells was used for quality control? Why not 16s rRNA from archaea and bacteria? Is this method directly targeting AMR genes in bacteria in the samples?

Use of the 18S rRNA gene as quality control provided the additional detection of human and fungal DNA. This assisted in validating low biomass samples – if the 18S rRNA gene was detected it is likely the sample was adequate, even if no bacterial DNA (as would be detected through 16S rRNA gene) was present. This assay is included as standard during manufacture by ThermoFisher as part of quality control. Future iterations of the AMR gene array could include a 16S rRNA gene target as suggested by the reviewer if space is made available.

There is potential that the assays on this card could be integrated within the array of a diagnostic array also developed by this research team (https://doi.org/10.1186/s13054-023-04303-1) which also integrated an 18S rRNA gene validation.

  1. The methods provided in the main text were inadequate. All must be moved from the supplementary file to the main text.

Methods moved to the main text as per recommendation.

  1. If the microbiology-based technique was also processed. Kindly revise as above mentioned.

The microbiological methods have been moved into the main manuscript as suggested.

Round 2

Reviewer 1 Report

Comments and Suggestions for Authors

the paper has been revised according to the comments and can be published